# Learning to Plan Optimistically: Uncertainty-Guided Deep Exploration via Latent Model Ensembles

**Tim Seyde**[*]  **Wilko Schwarting**[*]  **Sertac Karaman**  **Daniela Rus**
MIT CSAIL  MIT CSAIL  MIT LIDS  MIT CSAIL

**Abstract:** Learning complex robot behaviors through interaction requires structured exploration. Planning should target interactions with the potential to optimize long-term performance, while only reducing uncertainty where conducive to this objective. This paper presents Latent Optimistic Value Exploration (LOVE), a strategy that enables deep exploration through optimism in the face of uncertain long-term rewards. We combine latent world models with value function estimation to predict infinite-horizon returns and recover associated uncertainty via ensembling. The policy is then trained on an upper confidence bound (UCB) objective to identify and select the interactions most promising to improve long-term performance. We apply LOVE to visual robot control tasks in continuous action spaces and demonstrate on average more than 20% improved sample efficiency in comparison to state-of-the-art and other exploration objectives. In sparse and hard to explore environments we achieve an average improvement of over 30%.

**Keywords:** Learning Control, Sample Efficiency, Exploration

## 1  Introduction

The ability to learn complex behaviors through interaction will enable the autonomous deployment of various robotic systems in the real world. Reinforcement learning (RL) provides a key framework for realizing these capabilities, but efficiency of the learning process remains a prevalent concern. Real-life applications yield complex planning problems due to high-dimensional environment states, which are further exacerbated by the agent's continuous action space. For RL to enable real-world autonomy, it therefore becomes crucial to determine efficient representations of the underlying planning problem, while formulating interaction strategies capable of exploring these representations efficiently.

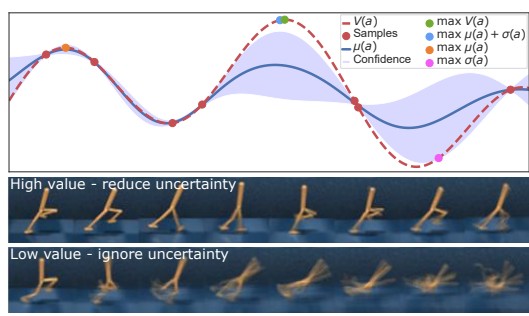

Figure 1: Top - Uncertainty over returns can guide exploration to regions with high potential for improvement (blue dot) during sampling-based value learning (red line). Bottom - uncertainty reduction should focus on expected high-reward behaviors.

In traditional controls, planning problems are commonly formulated based on the underlying state-space representation. This may inhibit efficient learning when the environment states are high-dimensional or their dynamics are susceptible to non-smooth events such as singularities and discontinuities [1, 2, 3]. It may then be desirable for the agent to abstract a latent representation that facilitates efficient learning [4, 5, 6]. The latent representation may then be leveraged either in a model-free or model-based setting. Model-free techniques estimate state-values directly from observed data to distill a policy mapping. Model-based techniques learn an explicit representation of the environment that is leveraged in generating fictitious interactions and enable policy learning in imagination [7]. While the former reduces potential sources of bias, the latter offers a structured representation encoding deeper insights into underlying environment behavior.

---

[*]Equal contribution. Correspondence to {`tseyde,wilkos`}@mit.edu

5th Conference on Robot Learning (CoRL 2021), London, UK.

The agent should leverage the chosen representation to efficiently identify and explore informative interactions. For example, an engineer would ideally only provide sparse high-level feedback to a robot and the system should learn without getting distracted by environment behavior irrelevant to task completion. This requires models that can disentangle behavioral surprise from expected performance and motivate the agent to investigate the latter. Figure 1 provides a one-dimensional example of a value mapping. The true function and its samples are visualized in red with the true maximum denoted by the green dot. Relying only on the predicted mean can bias policy learning towards local optima (orange dot; Sutton and Barto [8]), while added stochasticity can waste samples on uninformative interactions. Auxiliary information-gain objectives integrate predicted uncertainty, however, uncertain environment behavior does not equate to potential for improvement (pink dot). It is desirable to focus exploration on interactions that harbor potential for improving overall performance. Combining mean performance estimates with their uncertainty into an upper confidence bound (UCB) objective provides a concise method for this (blue dot; Auer et al. [9], Krause and Ong [10]). The uncertainty can be explicitly represented by an ensemble of hypothesis on environment behavior [11, 12]. Figure 1 demonstrates this selective uncertainty reduction by providing forward predictions of a model ensemble on two motion patterns for a Walker. The expected high-reward walking behavior has been sufficiently explored and the models strongly agree, while little effort has been extended to reduce uncertainty and learn the specific details of the expected low-reward falling behavior.

This paper demonstrates that exploring interactions through imagined positive futures can yield information-dense sampling and data-efficient learning. We present latent optimistic value exploration (LOVE), an algorithm that leverages optimism in the face of uncertain long-term rewards in guiding exploration. Potential futures are imagined by an ensemble of latent variable models and their predicted infinite-horizon performance is obtained in combination with associated value function estimates. Training on a UCB objective over imagined futures yields a policy that behaves inherently optimistic and focuses the robot learner on interactions with the potential to improve performance under its current world model. This provides a concise, differentiable framework for driving deep exploration while not relying on stochasticity. LOVE therefore provides an agent with

- an uncertainty-aware model for diverse hypotheses on trajectory evolution and performance,
- intrinsic exploration without reward feedback by active querying of uncertain long-term returns,
- targeted exploration guided by improvement potential, ignoring uncertainty tangential to the task.

We present empirical results on challenging visual robot control tasks that highlight the necessity for deep exploration in scenarios with sparse rewards, where we achieve on average more than 30% improvement over state-of-the-art and an exploration baseline. We further demonstrate 15% average improvement on benchmarking environments from the DeepMind Control Suite [13]. We compare to the recent agents Dreamer [7], a curiosity-based agent inspired by Sekar et al. [14], and DrQ [15].

## 2 Related work

**Problem representation**   Model-free approaches learn a policy by directly estimating performance from interaction data. While their asymptotic performance previously came at the cost of sample complexity [16, 17, 18], recent advances in representation learning through contrastive methods and data augmentation have improved their efficiency [19, 20, 15]. However, their implicit representation of the world can make generalization of learned behaviors under changing task specifications difficult. Model-based techniques leverage a structured representation of their environment that enables them to imagine potential interactions. The nature of the problem hereby dictates model complexity, ranging from linear [21, 22], over Gaussian process models [23, 24] to deep neural networks [25, 26]. In high-dimensional environments, latent variable models can provide concise representations that improve efficiency of the learning process [27, 4, 6, 7].

**Planning interactions**   Model-based approaches leverage their representation of the world in predicting the performance of action sequences. The agent may then either solve a model predictive control-style optimization [28, 25, 29] or train a policy in simulation [30, 26]. The resulting finite-horizon formulations can be extended by value function estimates to approximate an infinite-horizon planning problem [31, 7, 32]. When considering learned models, ensembling the model predictions may further be leveraged in debiasing the actor during training [30, 25, 26, 32]. Both explicit and implicit model rollouts together with value estimation can accelerate model-free learning [33, 34, 35].

**Directed exploration**   Directed exploration can improve over random exploration by focusing on information-dense interactions [36]. These methods are commonly driven by uncertainty estimates. Information gain techniques define an auxiliary objective that encourages exploration of unexpected environment behavior or model disagreement and have been applied in discrete [37, 38, 39] and continuous actions spaces [40, 41, 42]. Improving knowledge of the dynamics yields general purpose models, while the agent may explore uncertain interactions tangential to a specific task objective. Alternatively, exploration can be driven by uncertainty over performance as encoded by value functions [11, 43, 44, 45, 46], multi-step imagined returns [47, 48] or their combination [31, 1, 32].

**Model-ensemble agents**   Related work on ensemble agents has demonstrated impressive results. We note key differences to our approach. ME-TRPO [30] leverages a dynamics ensemble to debias policy optimization on finite-horizon returns under a known reward function and random exploration. MAX [49] and Amos et al. [50] explore via finite-horizon uncertainty in a dynamics ensemble. Plan2Explore [14] combines this idea with Dreamer [7], learning a model that enables adaptation to multiple downstream tasks. RP1 [51] explores in reward space via finite-horizon returns, but assumes access to the nominal reward function and full proprioceptive feedback. Seyde et al. [32] leverage full proprioception and embed optimism into the value function, which prohibits adjustment of the exploration trade-off during learning and limits transferability. Concurrent work by Rafailov et al. [52] penalizes uncertainty over latent states for learning robust visual control policies via offline RL. Exploring uncertain dynamics samples interactions orthogonal to task completion and finite-horizon objectives limit exploration locally, while full-observability and access to the reward function are strong assumptions. We learn latent dynamics, reward and value functions under partial observability to explore uncertain infinite-horizon returns. This enables backpropagation through imagined trajectories to recover analytic policy gradients, as well as guided deep exploration based on expected potential for long-term improvement.

## 3   Preliminaries

### 3.1   Problem formulation

We formulate the underlying optimization problem as a partially observable Markov decision process (POMDP) defined by the tuple $\{\mathcal{X}, \mathcal{A}, T, R, \Omega, \mathcal{O}, \gamma\}$, where $\mathcal{X}, \mathcal{A}, \mathcal{O}$ denote the state, action and observation space, respectively, $T\colon \mathcal{X} \times \mathcal{A} \to \mathcal{X}$ signifies the transition mapping, $R\colon \mathcal{X} \times \mathcal{A} \to \mathbb{R}$ the reward mapping, $\Omega\colon \mathcal{X} \to \mathcal{O}$ the observation mapping, and $\gamma \in [0, 1)$ is the discount factor. We define $x_t$ and $a_t$ to be the state and action at time $t$, respectively, and use the notation $r_t = R(x_t, a_t)$. Let $\pi_\phi(a_t|o_t)$ denote a policy parameterized by $\phi$ and define the discounted infinite horizon return $G_t = \sum_{\tau=t}^{\infty} \gamma^{\tau-t} R(x_\tau, a_\tau)$, where $x_{t+1} \sim T(x_{t+1}|x_t, a_t)$ and $a_t \sim \pi_\phi(a_t|o_t)$. The goal is then to learn the optimal policy maximizing $G_t$ under unknown nominal dynamics and reward mappings.

### 3.2   Planning from pixels

We build on the world model introduced in Hafner et al. [29] and refined in Hafner et al. [7]. High-dimensional image observations are first embedded into a low-dimensional latent space using a neural network encoder. A recurrent state space model (RSSM) then provides probabilistic transitions and defines the model state $s$. Together, the encoder and RSSM define the representation model. The agent therefore abstracts observation $o_t$ of environment state $x_t$ into model state $s_t$, which is leveraged for planning. Consistency of the learned representations is enforced by minimizing the reconstruction error of a decoder network in the observation model and the ability to predict rewards of the reward model. For details, we refer the reader to Hafner et al. [7], and provide their definitions

$$
\begin{aligned}
&\text{Representation model:} &&p_\theta(s_t|s_{t-1}, a_{t-1}, o_t) \\
&\text{Transition model:} &&q_\theta(s_t|s_{t-1}, a_{t-1}) \\
&\text{Observation model:} &&q_\theta(o_t|s_t) \\
&\text{Reward model:} &&q_\theta(r_t|s_t),
\end{aligned}
\tag{1}
$$

where $p$ and $q$ denote distributions in latent space, with $\theta$ as their joint parameterization. The action model $\pi_\phi(a_t|s_t)$ is then trained to optimize the predicted return of imagined world model rollouts. The world model is only rolled-out over a finite horizon $H$, but complemented by predictions from

the value model $v_\psi(s_t)$ at the terminal state $s_{t+H}$ to approximate the infinite horizon return. The policy and value function are trained jointly using policy iteration on the objective functions

$$\max_\phi E_{q_\theta, q_\phi} \left( \sum_{\tau=t}^{t+H} V_\lambda(s_\tau) \right), \qquad \min_\psi E_{q_\theta, q_\phi} \left( \sum_{\tau=t}^{t+H} \frac{1}{2} \|v_\psi(s_\tau) - V_\lambda(s_\tau)\|^2 \right), \qquad (2)$$

respectively. Here, $V_\lambda(s_\tau)$ represents an exponentially recency-weighted average of the $k$-step value estimates $V_N^k(s_\tau)$ along the trajectory to stabilize the learning [8] as further specified in Appendix K.

## 4 Uncertainty-guided latent exploration

The world model introduced in Section 3.2 can be leveraged in generating fictitious interactions for the policy to train on. However, the learned model will exhibit bias in uncertain regions where insufficient samples are available. Training on imagined model rollouts then propagates simulation bias into the policy. Here, we address model bias by ensembling our belief over environment behavior. We can leverage the underlying epistemic uncertainty in formulating a UCB objective for policy learning that focuses exploration on regions with high predicted potential for improvement.

### 4.1 Model learning with uncertainty estimation

The model parameters are only weakly constrained in regions where interaction data is scarce and random influences strongly impact prediction performance. In order to prevent the agent from learning to exploit these model mismatches, we consider predictions from an ensemble. Individual predictions will align in regions of high data support and diverge in regions of low support. The ensemble mean then serves as a debiased estimator of environment behavior, while the epistemic uncertainty is approximated via model disagreement [12]. We consider an ensemble of $M$ latent-space particles. Each particle is represented by a unique pairing of a transition, reward and value model to yield,

$$\text{Particles: } \{(q_{\theta_i}(s_t|s_{t-1}, a_{t-1}), \ q_{\theta_i}(r_t|s_t), \ v_{\psi_i}(s_t))\}_{i=1}^M. \qquad (3)$$

The encoder and decoder remain shared to ensure that all particles leverage the same latent space, while the internal transition dynamics retain the ability of expressing distinct hypothesis over environment behavior. For particle $i$, we define the predicted infinite-horizon trajectory return as

$$G_{t,i}(\theta_i, \psi_i, \phi) = \sum_{\tau=t}^{t+H} V_{\lambda,i}(s_\tau), \qquad (4)$$

where $V_{\lambda,i}(s_\tau)$ is computed via Eq. (8) with the particle's individual transition, reward and value models. Distinctness of the particles is encouraged by varying the initial network weights between ensemble members and shuffling the batch order during training. Predicted trajectory returns with corresponding uncertainty estimates are then obtained by considering the ensemble mean and variance,

$$\mu_G(\theta, \psi, \phi) = \frac{1}{M} \sum_{i=1}^M G_{t,i}(\theta_i, \psi_i, \phi), \qquad (5)$$

$$\sigma_G^2(\theta, \psi, \phi) = \frac{1}{M} \sum_{i=1}^M (G_{t,i}(\theta_i, \psi_i, \phi) - \mu_G(\theta, \psi, \phi))^2. \qquad (6)$$

### 4.2 Policy learning with directed exploration

The policy learning objective in Eq. (2) could be replaced by the ensemble mean in Eq. (6). This would reduce model bias in the policy, but require an auxiliary objective to ensure sufficient exploration. We consider exploration to be desirable when it reduces uncertainty over realizable task performance. The trajectory return variance in Eq. (6) encodes uncertainty over long-term performance of actions. In combination with the expected mean, we recover estimated performance bounds. During data acquisition, we explicitly leverage the epistemic uncertainty in identifying interactions with potential for improvement and define the acquisition policy objective via an upper confidence bound (UCB) as

$$G_{aq}(\theta, \psi, \phi) = \mu_G(\theta, \psi, \phi) + \beta \sigma_G(\theta, \psi, \phi), \qquad (7)$$

where the scalar variable $\beta$ quantifies the exploration-exploitation trade-off. For $\beta < 0$ we recover a safe-interaction objective, while $\beta > 0$ translates to an inherent optimism that uncertainty harbors potential for improvement. Here, we learn an optimistic policy $\pi_{\phi_{aq}}$ that is intrinsically capable of deep exploration and focuses interactions on regions with high information-density. Furthermore, in the absence of dense reward signals, the acquisition policy can leverage prediction uncertainty in driving exploration. This behavior is not limited to the preview window, as the value function ensemble projects long-term uncertainty into the finite-horizon model rollouts. While training in imagination, we leverage the optimistic policy to update our belief in regions that the acquisition policy intends to visit. In parallel, we train an evaluation policy $\pi_{\phi_{ev}}$ that aims to select the optimal actions under our current belief. The evaluation policy optimizes the expected mean return ($\beta = 0$).

### 4.3 Latent optimistic value exploration (LOVE)

In the following, we provide a high-level description of the proposed algorithm, LOVE, together with implementation details of the overall training process and the associated pseudo-code in Algorithm 1.

**Summary** LOVE leverages an ensemble of latent variable models in combination with value function estimates to predict infinite-horizon trajectory performance and associated uncertainty. The acquisition policy is trained on a UCB objective to imagine positive futures and focus exploration on interactions with high predicted potential for long-term improvement. The ensemble members are constrained to operate over the same latent space to encourage learning of abstract representations conducive to the objective, while ensuring prediction consistency.

**Implementation** The algorithm proceeds in two alternating phases. In the online phase, the agent leverages its acquisition policy to explore interactions optimistically and resulting transitions are appended to memory $\mathcal{D}$. In the offline phase, the agent first updates its belief about environment behavior and then adjusts its policy accordingly. The represen-

---

**Algorithm 1:** LOVE

**Initialize :** random parameters $\{\theta_i, \psi_i, \phi_{aq}, \phi_{ev}\}$
**for** *episode in episodes* **do**
  **for** *timestep $t = 1$ to $T$* **do**
    $a_t \sim \pi_\phi(\cdot|s_t)$, $s_t \sim p_{\theta_i}(\cdot|s_{t-1}, a_{t-1}, o_t)$
    Observe environment transition and add to $\mathcal{D}$
  **for** *trainstep $s = 1$ to $S$* **do**
    **for** *particle $i = 1$ to $M$* **do**
      Sample sequence batch
      $\{(o_t, a_t, r_t)\}_{t=b}^{b+L} \sim D$
      Compute states $s_t \sim p_{\theta_i}(s_t|s_{t-1}, a_{t-1}, o_t)$
      Estimate values $V_{\lambda,i}(s_\tau) \leftarrow \mathbf{rollout}(s_t, i)$
      Representation learning of $\theta_i$ on
      $\{(r_t, o_{t+1})\}_{t=b}^{b+L}$
      Regression of $\psi_i$ on $V_{\lambda,i}(s_\tau)$ targets in (2)
    Sample sequences $\{(o_t, a_t, r_t)\}_{t=b}^{b+L} \sim D$
    **for** *particle $i = 1$ to $M$* **do**
      Compute states $s_t \sim p_{\theta_i}(s_t|s_{t-1}, a_{t-1}, o_t)$
      Estimate values $V_{\lambda,i}(s_\tau) \leftarrow \mathbf{rollout}(s_t, i)$
    Compute ensemble statistics $\mu_G, \sigma_G$ via (6)
    Update $\phi_{aq}$ by optimizing UCB objective in (7)
    Update $\phi_{ev}$ by optimizing mean returns in (5)

---

tation learning step extends the procedure introduced in Hafner et al. [7] to ensemble learning and groups each model with a unique value function estimator into a particle. The batch order is varied between particles during training to ensure variability in the gradient updates and to prevent collapse into a single mode. The policy learning step combines particle predictions to generate value targets in Eq. (4) by simulating ensemble rollouts from the same initial conditions. The trajectory return statistics of Eq. (6) and Eq. (5) are combined into the UCB objective of Eq. (7), which the acquisition policy optimizes, while the evaluation policy optimizes the mean return in Eq. (5).

## 5 Experiments

We evaluate LOVE on continuous visual control tasks. First, we showcase intrinsic exploration in the absence of reward signals. Then, we illustrate interpretable uncertainty estimation of long-term returns with sparse reward signals. Finally, we benchmark performance on tasks from the DeepMind Control Suite [13]. We use a single set of hyperparameters throughout, as detailed in Appendix A.

### 5.1 Intrinsic exploration without reward feedback

In the absence of reward feedback an agent relies on intrinsic motivation to explore. LOVE enables intrinsic exploration by considering uncertainty over infinite-horizon returns. Without informative mean performance estimates, optimism acts as a proxy to guide exploration (see Eq. 7).

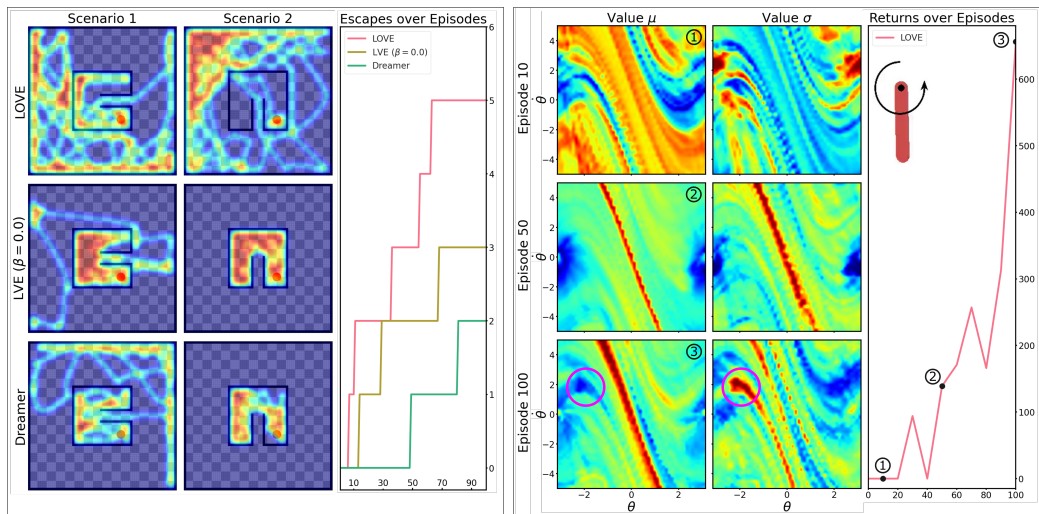

Figure 2: **Left** - the agent starts inside a bug trap (red dot) and does not receive reward feedback. We provide occupancy maps for LOVE, LVE (no optimism), and Dreamer with escape counts on 3 seeds. LOVE's ability to consider uncertain long-term performance enables consistent exploration in the absence of reward signals yielding the highest escape rate and area coverage. **Right** - Pendulum initialized at the bottom with sparse rewards at the top. Predicted ensemble mean (left) and standard deviation (right) of the value function. The agent first learns to quantify model uncertainty (Ep. 10), and then discovers high reward potential at the top (Ep. 50). Once the agent learned to reach the top, remaining uncertainty with low potential for improvement is not further explored (Ep. 100, circle).

We consider a reward-free bug trap environment. The agent starts inside the trap and can exit through a narrow passage. There is no reward feedback to guide planning and escapes need to arise implicitly through exploration. The agent is represented as a pointmass under continuous actuation-limited acceleration control and observes a top-down view of the environment. The ratio of agent diameter to passage width is $0.9$ and collisions are modelled as inelastic with the coefficient of restitution set to $0.3$. The relative size constrains the set of policies allowing for entry into the passage, while inelastic collisions reduce random motions.

We compare performance of LOVE to both Dreamer and LVE, a variation that ablates on LOVE's optimism ($\beta = 0$). Two variations of the environment are run on 3 random seeds for 100k timesteps. Exemplary occupancy maps are provided in Figure 2 (left). We note that LOVE escapes the bug trap in both scenarios to explore a much larger fraction of the state space than either LVE or Dreamer. This can be attributed to LOVE's ability to envision uncertain rewards beyond the preview horizon, which optimistically drives deep exploration in the absence of reward signals (row 1). Removing long-term optimism by only guiding interactions through mean performance estimates can lead to prediction collapse and the agent assuming the absence of reward without explicitly querying the environment for confirmation (row 2). We observe similar behavior for the Dreamer agent, which employs a single latent model and leverages random noise for exploration (row 3). The total escape count confirms this insight (column 3), where the remaining occupancy maps are provided in Appendix E.

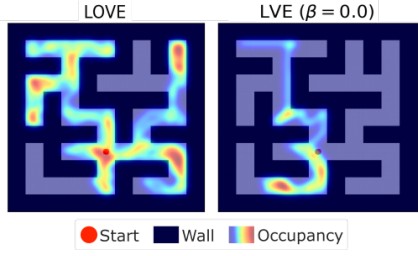

Figure 3: Reward-free maze exploration.

We further highlight the differences in intrinsic exploration between LOVE and LVE on a reward-free maze exploration task. The environment follows the same physics as the bug trap and the ratio of agent diameter to corridor width is $0.33$. The occupancy maps in Figure 3 exemplify how optimism over returns can set its own exploration goals in domains without reward feedback. These observations further underline the potential of combining model ensembling for policy debiasing with optimism over infinite-horizon returns in guiding deep exploration.

## 5.2 Targeted exploration with sparse rewards

Efficient planning under sparse reward feedback requires long-term coordination. LOVE focuses on interactions with high potential for improving infinite-horizon performance by trading-off predicted returns with their uncertainty. The agent then only reduces uncertainty when it aligns with successful task completion. We consider a sparse reward pendulum task, as its functional representations are readily interpretable. The pendulum is always initialized in a downward configuration and only receives reward at the top. The agent can therefore not rely on random initialization to observe initial reward signals and needs to actively explore the upright configuration.

Figure 2 (right) visualizes the mean and variance of the value ensemble during training. We generate predictions in state-space by providing 5 contextual images for velocity estimation. Initially, the agent builds a world model within which it learns to quantify uncertainty (Ep. 10). The agent then leverages the model to explore the top. This yields a spinning behavior and reward propagates into the value mean, while uncertainty is allocated for trajectories that could yield stabilization at the top (Ep. 50). The agent queries these promising motions and refines its behavior to reach the upright, reducing associated uncertainty (Ep. 100). The UCB objective allows the agent to ignore remaining uncertainty tangential to successful task completion (circle). LOVE therefore guides exploration via potential for long-term improvement while offering interpretable uncertainty estimates.

## 5.3 Performance with mixed sparse and dense rewards

The previous sections highlighted LOVE's intrinsic motivation to explore potential for long-term improvement based on its uncertainty estimation over returns. In the following, we compare performance on a variety of visual control tasks from the DeepMind Control Suite and sparsified variations.

The environments feature $(64, 64)$ image observations, continuous actions, and dense or sparse rewards. The sparse Cheetah Run, Walker Run and Walker Walk tasks are generated by zeroing rewards below a threshold ($0.25$, $0.25$ and $0.7$, respectively; rescaled to $[0, 1]$). We use the same set of parameters throughout all experiments mirroring those of Dreamer. However, we do not use random exploration noise. We furthermore use an ensemble of $M = 5$ latent models, an initial UCB trade-off of $\beta = 0.0$, an episodic growth of $\delta = 10^{-3}$, and alter the policy learning rate and training steps to account for the more complex UCB objective. The changes to the policy learning rate and training steps are also applied to Dreamer to yield $\Delta$Dreamer, as we found this to improve performance (see Appendix G). We furthermore introduce an exploration baseline, $\Delta$Dreamer+Curious, inspired by the curiosity bonus in Sekar et al. [14] over an ensemble of RSSM dynamics.

Table 1 provides benchmarking performance on 5 seeds over 300 episodes (see also Appendix F). On the sparsified environments, LOVE achieves $40\%$ average improvement over the state-of-the-art $\Delta$Dreamer agent and $30\%$ over both the exploration baseline $\Delta$Dreamer+Curious and ensembling baseline LVE. On the other environments, LOVE improves by $26\%$, $37\%$, and $9\%$, respectively. This suggests that LOVE's combination of latent model ensembling with directed exploration can aid in identifying interactions conducive to task completion. Ensembling reduces propagation of model-specific biases into the policy. Optimistic exploration of uncertain long-term returns focuses sampling on regions with promising performance estimates while ignoring uncertain regions that are tangential to task completion. This is an advantage of formulating the exploration objective in terms of uncertain rewards and not uncertain dynamics, as shown by the comparison to $\Delta$Dreamer+Curious.

We ablate the performance of LOVE on the UCB trade-off parameter $\beta$ and compare against LVE ($\beta = 0$). We note that while LOVE outperforms LVE on the majority of tasks, both reach similar terminal performance in several instances. However, LOVE provides clear improvements on the fully-sparse Cartpole Swingup and the partially-sparse Hopper Hop tasks, as well as the sparse locomotion tasks (Table 1, top). These environments initialize the agent in configurations that provide no reward feedback, thereby forcing the agent to actively explore. LOVE leverages uncertainty-guided exploration and gains an advantage under these conditions. Similarly, this can explain performance on the Pendulum Swingup task. While the task only provides sparse reward feedback, random environment initializations offer sufficient visitation to the non-zero reward states, removing the need for active exploration. LOVE also improves performance on the dense Walker tasks. These environments require learning of stable locomotion patterns under complex dynamics, where directed exploration efficiently identifies tweaks to the gait. Similar to the bug trap environment of Section 5.1,

| 300k steps | LOVE | LVE | ΔDreamer | +Curious | DrQ | A3C | D4PG |
|---|---|---|---|---|---|---|---|
| Cheetah Sparse (R) | $614_{\pm262}$ | $606_{\pm117}$ | $259_{\pm280}$ | $514_{\pm302}$ | – | – | – |
| Walker Sparse (R) | $234_{\pm129}$ | $164_{\pm151}$ | $78_{\pm60}$ | $106_{\pm120}$ | – | – | – |
| Walker Sparse (W) | $935_{\pm24}$ | $864_{\pm135}$ | $584_{\pm315}$ | $639_{\pm277}$ | – | – | – |
| Hopper Sparse (S) | $831_{\pm72}$ | $778_{\pm142}$ | $649_{\pm212}$ | $530_{\pm246}$ | – | – | – |
| Avg. Diff. to LOVE | +0% | -11% | -46% | -35% | – | – | – |
| Cartpole Dense | $688_{\pm124}$ | $666_{\pm78}$ | $512_{\pm253}$ | $539_{\pm101}$ | $781_{\pm100}$ | $558_{\pm7}$ | $862_{\pm1}$ |
| Cartpole Sparse | $631_{\pm259}$ | $192_{\pm132}$ | $388_{\pm245}$ | $219_{\pm199}$ | $231_{\pm337}$ | $180_{\pm6}$ | $482_{\pm57}$ |
| Cheetah Run | $771_{\pm118}$ | $753_{\pm120}$ | $655_{\pm93}$ | $760_{\pm73}$ | $533_{\pm143}$ | $214_{\pm2}$ | $524_{\pm7}$ |
| Finger Spin | $584_{\pm302}$ | $605_{\pm306}$ | $348_{\pm156}$ | $380_{\pm173}$ | $898_{\pm131}$ | $129_{\pm2}$ | $986_{\pm1}$ |
| Hopper Hop | $203_{\pm104}$ | $165_{\pm96}$ | $118_{\pm95}$ | $83_{\pm87}$ | $151_{\pm51}$ | $1_{\pm0}$ | $242_{\pm2}$ |
| Pendulum | $641_{\pm298}$ | $639_{\pm332}$ | $320_{\pm353}$ | $177_{\pm208}$ | $399_{\pm298}$ | $49_{\pm5}$ | $681_{\pm42}$ |
| Walker Run | $528_{\pm123}$ | $506_{\pm67}$ | $438_{\pm85}$ | $451_{\pm119}$ | $338_{\pm82}$ | $192_{\pm2}$ | $567_{\pm19}$ |
| Walker Walk | $947_{\pm36}$ | $914_{\pm52}$ | $902_{\pm94}$ | $873_{\pm122}$ | $815_{\pm184}$ | $311_{\pm2}$ | $968_{\pm2}$ |
| Avg. Diff. to LOVE | +0% | -12% | -29% | -35% | -18% | -70% | +9% |

Table 1: DeepMind Control Suite. Mean and standard deviation on 9 seeds after $3 \times 10^5$ timesteps. LOVE improves sample efficiency particularly under sparse reward feedback. Temporally-extended optimism helps LOVE in actively exploring uncertain returns, providing an advantage over LVE. Formulating intrinsic motivation in reward-space enables LOVE to identify uncertain interactions conducive to solving the task, providing an advantage over the curiosity baseline ΔDreamer+Curious. *D4PG, A3C: converged results at $10^8$ timesteps for reference.

we observe that optimistic exploration is especially favoured by objectives that provide sparse reward feedback, while enabling efficient discovery of tweaks to motion patterns under complex dynamics.

We additionally compare to Data Regularized Q (DrQ), a concurrent model-free approach. DrQ updates its actor-critic models online giving it an advantage over LOVE and Dreamer. LOVE performs favourably on the majority of environments with significant differences on the sparse tasks (Pendulum, Cartpole Sparse) and the locomotion tasks (Hopper, Walker). DrQ outperforms LOVE on Finger Spin and Cartpole with dense rewards. On these tasks, learning an explicit world model may actually be detrimental to attaining performance quickly. The former task features high-frequency behaviors that may induce aliasing, while the latter task allows the agent to leave the frame yielding transitions with no visual feedback. We additionally provide converged performance results for pixel-based D4PG [53] and proprioception-based A3C [54] at $10^8$ environment steps to put the results into perspective.

## 6   Conclusion

We propose Latent Optimistic Value Exploration (LOVE), a model-based reinforcement learning algorithm that leverages long-term optimism to guide exploration for continuous visual control. LOVE leverages finite-horizon rollouts of a latent model ensemble in combination with value function estimates to predict long-term performance of candidate action sequences. The ensemble predictions are then combined into an upper confidence bound objective for policy learning. Training on this objective yields a policy that optimistically explores interactions that have the potential of improving task performance while ignoring uncertain interactions tangential to task completion. These aspects are particularly important for robot control in complex environments, where the engineer aims to provide sparse high-level feedback without intensive reward engineering and the system should efficiently learn without getting distracted by irrelevant environment behavior. To this end, we evaluate LOVE regarding its exploration capabilities and performance on a variety of tasks. In the absence of reward signals, LOVE demonstrates an intrinsic motivation to explore interactions based on their information density. Empirical results on DeepMind Control Suite tasks showcase LOVE's competitive performance and ability to focus exploration on interactions conducive to task completion, particularly when reward signals are sparse. Lifting intrinsic motivation into reward-space is then preferred over dynamics-space, as shown by comparison to a curiosity baseline. Furthermore, LOVE demonstrates improved sample efficiency over the recent model-based Dreamer agent and competitiveness with the model-free DrQ agent.

**Acknowledgements**

This work was supported in part by the Office of Naval Research (ONR) Grant N00014-18-1-2830, Qualcomm and Toyota Research Institute (TRI). This article solely reflects the opinions and conclusions of its authors and not TRI, Toyota, or any other entity. We thank them for their support. The authors further would like to thank Lucas Liebenwein for assistance with cluster deployment, and acknowledge the MIT SuperCloud and Lincoln Laboratory Supercomputing Center for providing HPC resources. We would also like to thank the reviewers and program chairs for their helpful feedback and suggestions for improvement.

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
