# OpenReview forum: "Learning to Plan Optimistically: Uncertainty-Guided Deep Exploration via Latent Model Ensembles"
_robot-learning.org/CoRL/2021/Conference — CoRL2021 Poster_

### Official Review · Reviewer_Z9mA · 2021-07-22

**Originality:** Good
**Technical Quality:** Good
**Clarity Of Presentation:** Very Good
**Impact:** 2

**Recommendation:**

Weak Accept: I recommend accepting the paper, but will not argue for my recommendation if the majority of other reviewers have a different opinion.

**Summary:**

This paper presents Latent Optimistic Value Exploration (LOVE), a learning method for visual robot control that leverages optimistic exploration. At its core, the method uses an ensemble of latent-space models that share a common encoder but different models for transitions, rewards and values. The different returns predicted by the ensemble members are then used to quantify uncertainty over the long term performance of actions, which in turn is used to define an optimistic learning objective based on UCB that encourages the policy to explore uncertain regions in the latent state space that likely yield good task performance. The method is compared against baselines on a set of scenarios from the DeepMind Control Suite where it shows encouraging results.

**Issues:**

- Please provide a study with a varying number of ensemble members (instead of just 2 and 5) to give the reader an intuition how important accurate uncertainty estimates are.

**Reviewer Expertise:**

Good: General knowledge of the area

**Strengths And Weaknesses:**

First, the paper is in general well written an easy to follow. The method mostly builds upon previous work on recurrent state space models (Haffner et al. '19) but from what I can tell, using an ensemble of latent-space models to formulate a UCB-type learning objective is new. I believe that the idea is sound, since using objectives for optimistic exploration that use some form of UCB have been very successful in tangential fields, including Bayesian RL or online POMDP planning.

Once concern I have regarding the proposed method is the relatively small (5) number of ensemble members. While I suppose that this number was kept small to reduce the number of parameters that have to be trained, it is not clear to me how estimates from such a small number yield accurate estimates for the mean and -- in particular -- the variance in eq.(5) and eq.(6). As a consequence, I'm not too convinced that the policy is actually encouraged to explore regions with uncertain optimistic returns. This also raises the question whether the reported performance benefit of LOVE compared to Dreamer really stems from a more efficient exploration of the environments or if there's something else in the architecture of LOVE that explains the results. While the ablation study in Appendix J suggests that an ensemble member size of 5 has an advantage over a size of 2, I would have liked to see a more extensive study with varying numbers of members (for instance 1 to, say 20). This would not only show how important more accurate uncertainty estimates are (if at all), it would also shed some light on the tradeoff between accuracy and parameter size.


One additional comment: In what sense is LOVE a "deep exploration" method?

**Summary Of Recommendation:**

The paper provides an interesting extension for on recurrent state space models and I believe the paper could have some impact in case some additional experiments are provided.

---

> ### Author Response · Authors · 2021-08-31
> **Response to Reviewer Z9mA**
>
> We would like to thank the reviewer for their time and insightful comments. We are excited that they appreciate the promise of applying optimistic UCB-type exploration to policy learning within learned latent world models.
>
> We have updated the manuscript with additional experiments (Appendices L, M, N, O) and increased the number of seeds for all agents in the Control suite evaluations (Table 1, Figure 9) from 5 to 9 for improved statistical significance.
>
> **In what sense is LOVE a "deep exploration" method?**
> We consider LOVE to leverage “deep exploration” as it’s policy learning objective estimates uncertainty over infinite-horizon returns. Each ensemble member features a value function that projects expected long-term behavior into the finite planning horizon, and through the UCB objective we recover uncertainty over long-term behavior. Many related works limit policy optimization to finite horizon rewards and therefore only consider short-term or local behavior (e.g. Section 2, “Model-ensemble agents” - ME-TRPO, MAX, RP1).
>
> **Please provide a study with a varying number of ensemble members (instead of just 2 and 5).**
> The model size can indeed become a limiting factor for large model ensembles. Generally, we found the common literature choice of 5 particles as used in ME-TRPO or RP1 to work sufficiently well, while e.g. MAX relied on only 3 models. We evaluate performance of an ensemble with 10 particles and consider the Cartpole Sparse, Cheetah Run, Pendulum Swingup and Walker Walk tasks to accommodate other additional experiments. We have added the data to Figure 13. Generally, increasing the ensemble size may improve performance slightly at the cost of excess computational burden particularly when considering GPU memory constraints. The common choice of 5 ensemble members may then yield a performant compromise.
>
> We thank the reviewer for their time and kind feedback, and hope that we were able to address open questions. We invite the reviewer to reconsider our submission based on the additional discussion, updated manuscript and novel experiments.

---

### Official Review · Reviewer_5s1s · 2021-07-24

**Originality:** Good
**Technical Quality:** Very Good
**Clarity Of Presentation:** Excellent
**Impact:** 4

**Recommendation:**

Strong Accept: I recommend accepting the paper and will argue for my recommendation even if other reviewers hold a different opinion.

**Summary:**

This paper extends latent-space model-based RL methods by incorporating ensembles for the latent model - modeling uncertainty in the latent transition model, reward function, and value function. These ensembles are factored into the overall policy iteration approach by taking an optimistic viewpoint on latent model uncertainty. To compute the policy learning objective, rollouts are performed for each ensemble member and used to estimate the infinite horizon return of the trajectory. These estimates are aggregated to form a mean and standard deviation, which are used to create an upper confidence bound. By training the policy on this upper confidence bound, the policy in inherently incentivized to visit states with high uncertainty if those states hold promise towards the desired task.

The authors demonstrate the approach on several simulated examples in the DeepMind control suite, comparing against several ablations and baselines.

**Issues:**

The authors should add a discussion of how hyperparameters were optimized and how sensitive the approach is to different hyperparameters such as the UCB coefficient growth rate.

**Reviewer Expertise:**

Good: General knowledge of the area

**Strengths And Weaknesses:**

Overall, this paper presents a clean approach to implementing optimism under uncertainty for exploration in latent model-based RL. The idea is clean, and the implementation through ensembles is simple, but effective. The paper is well written and easy to follow. The experiments were thorough, and I appreciated the insightful qualitative results showing how the approach leads to deep exploration in the bug-trap environments, and that the exploration is targeted, through the snapshots of uncertainty in the cart pole experiment. I also appreciated the visualizations of the dynamics model predictions, showing the uncertainty reduction around high-reward behavior while no uncertainty reduction on low-reward behaviors. The quantitative results were also convincing, and I appreciated that the authors included model-free baselines, even if these baselines sometimes performed better than the latent model-based approach.

The paper could be improved with more details on the implementation of the ensembles, and discussion of hyperparameter optimization. While the policy optimization objective G was clearly defined, it was not clear how the policy was optimized. Furthermore, the success of LOVE depends significantly on the ensemble’s uncertainty being high enough to encourage the model to explore, rather than get stuck in a suboptimal solution. The authors handle this in the paper by increasing the UCB coefficient at every episode in an ad-hoc manner, as opposed to using a calibrated confidence bound as suggested in the original UCB work in the bandits literature. More discussion on the impact of hyperparameters like the growth rate \delta would strengthen this paper.


**Summary Of Recommendation:**

Overall, while the idea in this paper is relatively simple, the paper has strong experimental results and is very well written, leading to my vote for acceptance.

---

> ### Author Response · Authors · 2021-08-31
> **Response to Reviewer 5s1s**
>
> We would like to thank the reviewer for their time and positive feedback. We are thrilled about their appreciation of research on combining model-based RL for visual control with targeted deep exploration.
>
> We have updated the manuscript with additional experiments (Appendices L, M, N, O) and increased the number of seeds for all agents in the Control suite evaluations (Table 1, Figure 9) from 5 to 9 for improved statistical significance.
>
> **While the policy optimization objective G was clearly defined, it was not clear how the policy was optimized.**
> The policy objective G in (7) is actually fully-differentiable as all components are parameterized via neural networks. We may therefore take analytic gradients with respect to this objective and train the acquisition policy with the Adam optimizer (see Appendix A). In addition to the explorative acquisition policy, we also train an evaluation policy that optimizes the expected ensemble mean. It is trained similarly on the objective in (7) by setting beta=0. We have added the evaluation policy objective to Algorithm 1 and reworded the corresponding text for improved clarity.
>
> **The authors should add a discussion of how hyperparameters were optimized and how sensitive the approach is to different hyperparameters such as the UCB coefficient growth rate.**
> We decided on a linear growth schedule for the UCB parameter to recover a relatively straight-forward approach that could be readily applied across all environments. One may likely recover better performance via more sophisticated formulations, particularly when adapting them to each environment individually. For example, Figure 12 suggests that a negative initial UCB parameter value can yield improved performance on the Hopper Hop task. This task has posed a challenge for many state-of-the-art algorithms in part due to its complex dynamics. Observing that an initial “safe-RL” approach may increase performance is interesting. However, we aimed for a single parameter set throughout to mirror the baseline Dreamer agent. Generally, starting with a strong positive UCB parameter is undesirable as the networks will not have recovered meaningful representations or values to propagate early on and would amplify initial parameter noise. Our choice therefore attempts to mitigate unfounded optimism during initialization (initial value 0), while encouraging exploration throughout the course of training (linear increase). We have expanded the discussion in Appendix I to include this. Additional discussion of parameter choices is provided in Appendix A.
>
> We would like to thank the reviewer for their time and positive feedback, and hope that we were able to clarify remaining questions.

---

### Official Review · Reviewer_KfAh · 2021-07-24

**Originality:** Poor
**Technical Quality:** Good
**Clarity Of Presentation:** Good
**Impact:** 3

**Recommendation:**

Weak Accept: I recommend accepting the paper, but will not argue for my recommendation if the majority of other reviewers have a different opinion.

**Summary:**

Learn a value function with model based reinforcement learning, augmented with an ensemble of dynamics models which are used to imagine rollouts and provide a variance estimate on the future return prediction. This estimate directs exploration by learning a aggregation policy which acquires rollouts based on the sum of the estimated return, and a beta-regularized variance of the return. This focuses the agent on regions of high return complexity, based on the magnitude of beta. The ensemble of models are trained to reconstruct the (distribution over) current state from observation and last state, action and current observation, predict the next state from last state and action, and predict reward. The ensemble of models is trained by shuffling input batch order, and network weight initialization.


**Issues:**

Primarily, an increased discussion of how this method significantly differs from Planning to Explore via Self-Supervised World Models (Sekar et. al.)

Additional experiments ablating on the network initialization which is used for model variance

**Reviewer Expertise:**

Very good: Comprehensive knowledge of the area

**Strengths And Weaknesses:**

Strengths:
The experimental ablation over beta values provided a much clearer understanding of how much the imagined model rollouts provided benefit as opposed to the exploration method

The algorithm performs well with relatively simple augmentations to the classic model based reinforcement learning framework.

Dreamer is a state of the art baseline in the same area that performs many similar computations, making it a decently good experimental comparison

The paper describes much of the core concepts clearly, so that it is easy to envision how the system is created. One note is that for how important it is to the core algorithm, how the diversity in the world models is actually injected is somewhat terse.


Weaknesses:

The use of ensemble of agents in reinforcement learning for exploration should probably be better charactized with comparison with: Planning to Explore via Self-Supervised World Models (Sekar et. al.). This is refernced in the paper, but the similarities between this work and the given work appear to be closer than that of the compared baseline, Dreamer. A longer discussion of why this work, which shares much of the same formalism, and same core method (explore where the ensemble disagrees), would be useful, because this work appears to be doing almost the same thing as Sekar et. al.

Since diversity in the model ensembles is only enforced at initialization, this seems like it would have deleterious effects, especially in low-return environments. For example, one can imagine a scenario where the nature of the environment makes it so that the networks are biased towards treating much of the unknown regions with the same output, thus making unseen states provide no intrinsic reward. Perhaps a method that incorporates ensemble diversity explicitly would be more effective in these cases. Additionally, because it does not appear that the forward models capture the variance of the target outputs, this might cause natural stochasticity in the environment to be interpreted as high exploration regions, since the means might as a result vary greatly based on how the values are optimized.

Model ensembles can be highly hyperparameter dependent, especially because network initialization and batch order randomization are the primary modes of injecting variance into the model ensemble. While a single set of hyperparameters seemed to work well in this case, ablations over the network initialization schemes could be useful to determine exactly how sensitive it is.

Even though exploration is supposed to be primarily directed by the variance in the dynamics models, in reality it appears that much of the results are from having an optimistic agent. By comparison, the compared method Dreamer does not appear to have this, using random noise instead. However, optimism should not be difficult to add to a method like Dreamer, and it would be useful to know exactly how much optimism adds, and how much exploration is actually encouraged through the variance reward

In the navigation task, because the objective, at least when comparing with the baseline performance, is to get good coverage, a comparison with an exploration algorithm which has been shown to get good coverage in these kinds of environments, such as Explore, Discover and Learn: Unsupervised Discovery of State-Covering Skills (Campos et. al.) would have been preferred, and more informative as the domain is not one that is commonly used.

While the deepmind control suite provides some interesting kinematics tasks, experiments on domains which require more exploration, such as a robot arm interaction with multiple object in an environment, or even monetzuma's revenge, might have been more informative. This is because the deepmind control tasks tend to be dense reward (forcing a hacked method for creating reward sparcity), and the exploration generally requires a more dense search of the environment, rather than a directed search towards reward-generating regions of the state space. This would help clarify how much of the benefit in exploration comes from the proposed ensemble of experts.


**Summary Of Recommendation:**

I propose a weak accept/borderline, primarily because of the similarity with existing work, the limitations of the algorithm in how variance is added and maintained in the model ensemble, and the limitations of the experimental component in properly clarifying the level of improvement based on optimism, and based on the proposed variance based intrinsic reward. An evaluation task that better highlighted this would have been desirable.

---

> ### Author Response · Authors · 2021-08-31
> **Response to Reviewer KfAh**
>
> We would like to thank the reviewer for their time and suggestions for improvement. We are very happy that they acknowledge the simplicity and effectiveness of our proposed method. In the following, we will briefly summarize our approach and then address individual questions.
>
> We have updated the manuscript with additional experiments (Appendices L, M, N, O) and increased the number of seeds for all agents in the Control suite evaluations (Table 1, Figure 9) from 5 to 9 for improved statistical significance.
>
> **Summary of approach:**
> LOVE aims to enable sample-efficient RL by guiding exploration through optimism over long-term returns. To this end, LOVE maintains a particle ensemble where each particle consists of a dynamics, reward and value model. The exploration policy is then optimized on a UCB objective over the particle returns. The associated uncertainty arises from disagreement in the dynamics, reward and value functions, while the value functions project long-term uncertainty into the finite planning horizon. The exploration policy then aims to reduce uncertainty over long-term returns only where it appears promising to improve overall performance and not solely for the sake of uncertainty reduction.
>
> **An increased discussion of how this method significantly differs from Planning to Explore via Self-Supervised World Models (Sekar et. al.) [would be useful].**
> Plan2Explore uses finite-horizon disagreement over a latent dynamics ensemble as an intrinsic exploration objective. It does not consider extrinsic reward signals during exploration and aims to build an informative world model that can facilitate specialization to any downstream task. It’s exploration objective is therefore curiosity-inspired and aims to reduce finite-horizon dynamics uncertainty. LOVE estimates infinite-horizon return uncertainty by ensembling the dynamics, reward and value function. The value functions project uncertain long-term returns into the finite horizon used for policy optimization. The two approaches are complementary in the sense that Plan2Explore aims to learn a capable world model that can adapt to various down-stream tasks, while LOVE focuses exploration on interactions with high potential of improving performance on the current tasks. LOVE also leverages dynamics uncertainty as part of its intrinsic motivation and may project the effects of long-term dynamics uncertainty into the finite planning horizon via the learned value functions. Here, our Dreamer + Curious baseline follows the idea of Plan2Explore and considers uncertainty from a dynamics ensemble as intrinsic motivation on top of the extrinsic reward signals provided by the environment. We have expanded the related works section by a short characterization of Plan2Explore.
>
> **Additional experiments ablating on the network initialization which is used for model variance.**
> For our experiments we have used the standard network initializations as employed by the Dreamer agent. Specifically, the kernel initializers used the Glorot-Uniform scheme and the initial biases were set to zero. We provide a comparison of exchanging either initializer with a Variance Scaling initializer with scale factor 0.333. We limit ourselves to the Walker Walk task due to computational constraints. Based on Figure 16, we observe that either change slightly lowers convergence speed with the same asymptotic performance. However, more sophisticated initialization schemes such as network parameter regularization towards anchor values as suggested in [55] may improve performance.
>
> **A comparison with an exploration algorithm which has been shown to get good coverage in [navigation tasks], such as Explore, Discover and Learn: Unsupervised Discovery of State-Covering Skills (Campos et. al.) would have been preferred.**
> Thank you very much for pointing us towards this interesting work. We note two important differences between the task considered here and the ones from EDL, namely the control structure and the observation type. While we consider acceleration control based on high-dimensional image observations, EDL considers position control based on state observations. To provide a qualitative comparison of the skills EDL learns we re-built our Maze environment within their framework and provide results in Figure 17 of Appendix O. We observe qualitatively similar exploration patterns, while it should be noted that the observations and dynamics differ between the two frameworks.
>
> We thank the reviewer for their insights and helpful suggestions, and hope that we were able to address any open questions. We invite the reviewer to reconsider our submission based on the additional discussion and novel experiments.

---

> > ### Comment · Reviewer_KfAh · 2021-09-08
> > **Response to Authors**
> >
> > Thank you for your responses, I appreciate the forthrightness with which the concerns were addressed, and the additions to the paper provided valuable insight.

---

### Official Review · Reviewer_BJGj · 2021-07-25

**Originality:** Fair
**Technical Quality:** Very Good
**Clarity Of Presentation:** Good
**Impact:** 3

**Recommendation:**

Weak Accept: I recommend accepting the paper, but will not argue for my recommendation if the majority of other reviewers have a different opinion.

**Summary:**

This paper presents an exploration mechanism that optimistically explores the state space in search of interactions that increase long-term performance. This is done by designing a UCB objective that maximises the sum of the expected infinite horizon return and a second term that captures the weighted uncertainty of these returns. Setting this weight (\beta) greater than zero leads to an optimistic objective that explores to reduce uncertainty around high-performant regions and avoids reducing uncertainty around low-reward regions that might not be task relevant. This objective is used together with Dreamer, a popular model-based RL algorithm to explore efficiently in high-dimensional continuous control tasks with pixel observations. The approach builds on top of Dreamer by introducing an ensemble of latent dynamics models, reward and value predictors through which the expected value and its uncertainty are computed as the mean and variance of the ensemble predictions (which are in turn computed via latent imagined rollouts). This exploration objective is used to train a policy for exploration; this policy is executed in the environment and optimistically explores the space to find high-performant transitions which are stored in a replay buffer. These transitions are used to train the dynamics, reward and value models as well as an “exploitation” policy via the standard Dreamer objective that uses imagined value gradients.

Several experiments are conducted to test different parts of the approach. First, a control experiment is done to show the exploration capabilities of the proposed agent; this is done in a task-agnostic manner without use of rewards. The agent is able to cover significant parts of the state space in this setting and does better than Dreamer and a baseline that optimise just the expected return for exploration. Next, the method is applied on continuous control tasks with sparse and dense rewards from DMControl (and a toy pendulum example). The method performs just as well or slightly better than baselines including Dreamer (with additional modifications including better hyper parameters and a dynamics-dissimilarity based objective for exploration), DrQ and other model-free methods on the dense reward tasks and beats baselines on sparse reward settings. Several visualisations of the learned exploration objective and ablations to different parameters (including the tradeoff \beta, imagination horizon and ensemble size) are provided in the supplement.

**Issues:**

Some comments:
1. Add details on how the evaluation policy is trained (at least in appendix). I presume this is trained with the standard Dreamer objective.
2. Is \beta bounded? If not, why would it not look at uncertain parts later down the line once \beta becomes sufficiently large? Do you also have to decay it later in the exploration phase to get the agent to do well asymptotically?
3. Why are results cut off at 300k steps? As mentioned above, prior papers (DrQ, CURL etc.) provide 500k (and Dreamer provides 1e6). How does the asymptotic performance look like? Does it drop in performance later?
4. On simple point mass task it did well without rewards, what would it do on complex tasks? Some tests in control suite would be quite illuminating — would also be nice to see some videos of exploration behaviour on these tasks.
5. Instead of executing the exploration policy only on the actor is it possible to interleave the execution of the exploration and exploitation policy on the actor? Say with a fixed probability sample from the exploration policy and otherwise run the exploitation policy. This may improve data efficiency further.
6. The thresholds for converting the dense reward to a sparse reward for the DMControl tasks are quite low, were these tuned for LOVE to get good results?
7. The different particles in the ensemble share the latent transition model, reward and value model. Is there a single decoder that is shared across the particles? Please mention this in the paper.
8. NIT: The title of the paper says “learning to plan”, but there is no planning (in the traditional sense) in the presented approach. I would prefer if the title were changed to something else that reflects the presented approach better.
9. NIT: Fig.1, the orange dot is hard to distinguish, please change the color.

**Reviewer Expertise:**

Very good: Comprehensive knowledge of the area

**Strengths And Weaknesses:**

This paper proposes an objective for optimistic exploration, implements it on top of Dreamer, a MBRL algorithm, and tests it on several continuous control tasks in simulation. The objective focuses on uncertain returns and pushes the agent to explore high-value regions with high-uncertainty; any uncertainty is seen as a positive, thereby making the approach risk-taking. The method shows good performance on the tasks it is tested on and can lead to improvements in data efficiency compared to model-based and model-free baselines, especially for sparse reward tasks. A few comments:
1. The idea of using a UCB baseline that leverages uncertainty in values rather than uncertain dynamics is appealing as it naturally focuses exploration on task-relevant regions. There are pros and cons to this: on a given downstream task, task-relevant exploration methods like the one proposed can lead to faster learning, but if the goal is to generate data that works well across tasks then task agnostic exploration methods (such as with uncertain dynamics) might be needed. Additionally, the latter can also generate interesting contact-rich transitions across domains (as such transitions can be hard to predict); it is not clear if the proposed approach also has the capability to do this.
2. While there is a control experiment showing the capability of this method to operate without any rewards, i.e. in a task-agnostic manner, this intuitively seems to depend heavily on initialisation of the value function as there are no returns for the value function learning to bootstrap from, solely relying on initialisation variance to account for the uncertain value predictions on unvisited states. This is in contrast with methods that use uncertain dynamics or dynamics prediction error. It would be useful if more intuition is provided on the behaviour of this task-agnostic exploration setting and it’s relationship to other exploration methods such as those using dynamics models or approaches such as Random Network Distillation (RND). Additionally, it would be useful to see videos of the task-agnostic exploratory behaviour when applied to the DMControl tasks presented in the paper. This will add further insight.
3. The proposed approach needs an ensemble of value estimators to compute the exploration objective, and as such can be applied on top of any standard actor-critic approach with either a value or Q-function ensemble. Instead, the proposed implementation uses Dreamer, a model-based method for this value estimation. Is there any reason why this method was chosen as using an actor-critic approach could have made implementation significantly easier and would have had far less moving parts thereby allowing better understanding of the parts that matter.
4. In a similar vein, the proposed approach already has an ensemble of latent dynamics models. Is it possible to combine the proposed objective with dynamics uncertainty (through some form of weighting for example) to capture complementary behaviour?
5. A potential weakness of the proposed approach, especially when concerning real world applications, is the optimistic point of view. Risk-sensitive or Risk-aware methods can be a better fit for real world robotics problems as they can be conservative with respect to their action choices. Can the proposed approach be applied on real robots? What issues might arise with such an application and how can they be mitigated? A discussion on this would be enlightening.
6. While the approach shows good performance on the sparse tasks in DMControl, especially showing improvements in data efficiency early on, it would also have been useful to compare the asymptotic performance of these methods, especially at 500k env steps as has been recently shown in several related papers such as DrQ, CURL, etc. Does the approach lead to similar asymptotic performance compared to baselines? Does the exploration need to be turned off for this to happen? This needs to be discussed further. It would also be helpful to see additional results on harder DMControl tasks and/or tasks from a different benchmark such as OpenAI Gym or even Atari; this would add significantly more strength to the paper.

**Summary Of Recommendation:**

This paper presents an exploration objective based on return uncertainty and applies it to several continuous control tasks showing improvements in data efficiency. The approach is simple and appealing as it encourages exploration in regions where returns can be potentially high, thereby encouraging task relevant exploration. Overall the experiments show good promise but more analysis needs to be done to clearly tease out the strengths and weaknesses of the proposed approach, especially in comparison with other exploration methods. Additionally, it would have been easier for the approach to have been applied on top of a standard actor-critic setup, it is not clear why this was not done. Lastly, the experiments on the continuous control tasks seem half-finished (w.r.t num env steps) and further experimentation, also on other tasks if possible, can provide more strength to the paper. I would consider it a borderline reject as it is currently.

---

> ### Author Response · Authors · 2021-08-31
> **Response to Reviewer BJGj**
>
> We would like to thank the reviewer for their time as well as their thorough and insightful comments. We are excited that they find the presented method appealing due to its promise of achieving targeted and task relevant exploration. In the following, we will try to clarify open questions and address remaining concerns.
>
> **It would have been useful to compare the asymptotic performance of [the] methods.**
> We have added asymptotic results for several environments in Figure 14 in Appendix L. We observe that LOVE matches or surpasses performance of converged D4PG on almost all environments aside from the Finger Spin task, which is challenging for model-based agents. The results were obtained without modifying or bounding the original UCB trade-off schedule.
>
> **Is it possible to interleave the execution of the exploration and exploitation policy on the actor?**
> We evaluated this for a sampling probability of 0.5 on the Cheetah and Walker Walk task due to computational constraints and provide results in Figure 15 in Appendix M. Within the scope of our experiment we do not find this to improve performance as too much exploration may have been suppressed.
>
> **If the goal is to generate data that works well across tasks then task agnostic exploration methods might be needed. [...] it is not clear if the proposed approach also has the capability to do this.**
> We completely agree that task agnostic exploration can help learn general purpose models and have reworded part of the related works to better capture this perspective. While LOVE formulates its exploration objective in reward space to accelerate task specific learning it does maintain an ensemble of dynamics functions whose prediction disagreement propagates into the uncertain returns. If so desired, one may further explicitly add the uncertainty over the dynamics ensemble to the intrinsic reward signal as suggested.
>
> **Is there any reason why [Dreamer] was chosen [as the baseline method]?**
> We chose Dreamer as our baseline as its model-based nature allows for training the exploration policy in imagination, further adding to the sample-efficiency. Dreamer is the current state-of-the-art for model-based RL on high-dimensional image inputs.
>
> **Is there any reason why this [model-based] method was chosen as using an actor-critic approach could have made implementation significantly easier.**
> The model ensemble directly captures epistemic uncertainty which we leverage for a more accurate uncertainty estimate over the expected return in addition to the epistemic uncertainty over the value function ensemble alone.
>
> **A potential weakness of the proposed approach, especially when concerning real world applications, is the optimistic point of view.**
> The agent’s risk preference can be adjusted by modifying the UCB trade-off parameter. Here, we considered positive values to achieve optimistic exploration. However, setting the parameter to negative values would yield a risk averse agent. This preference can further be adjusted. For example, in Figure 12 we find that on the Hopper task initializing the agent pessimistically and transitioning to optimism improves performance. This is likely due to the task’s complex dynamics requiring the agent to first build a satisfactory dynamics model through safe interaction from which interactions can then be explored optimistically. This could be a desirable approach for complex real-world systems.
>
> **Add details on how the evaluation policy is trained.**
> The evaluation policy optimizes the ensemble mean return in Equation (5) (see also L184). For improved visibility, we have included it in Algorithm 1 as well as the associated description (additions in blue).
>
> **Is \beta bounded?**
> For simplicity we only employed the linear growth schedule for \beta, although one may consider more complex schedules. It is important to note that the \beta parameter is linked to the exploration policy and encodes its optimism when sampling new interactions. The evaluation policy does not consider \beta and simply optimizes the expected mean return over the collected samples.
>
> **How were the sparse reward threshold determined on the DMC tasks?**
> These were roughly empirically determined by inspecting training logs and increasing the threshold until almost only zero rewards were observed during the initial episodes. This was to prevent random motion patterns from generating dense reward traces.
>
> **Is there a single decoder that is shared across the particles?**
> Thank you for pointing this out, we have added this information to L160.
>
> We thank the reviewer for their time and valuable insights, and hope that we were able to clarify open questions and address remaining concerns. We invite the reviewer to reconsider our submission based on the additional discussion and experiments, and to potentially adjust their score.

---

> > ### Author Response · Authors · 2021-09-03
> > **Updated manuscript and response**
> >
> > Dear reviewer, thank you again for your time and valuable feedback.
> > As today is the last day for updating scores, please don't hesitate to ask if you would like us to elaborate on any aspects of the updated manuscript or the response.
> > Thanks!

---

> > > ### Comment · Reviewer_BJGj · 2021-09-04
> > > **Thanks for the rebuttal**
> > >
> > > I would like to thank the authors for their answers in the rebuttal and additional experiments in the supplementary material which helped clarify many of my concerns regarding the paper. I will increase my score to a "Weak Accept".

---

### Author Response · Authors · 2021-08-31
**Updated manuscript and main changes**

We would like to thank the reviewers and the AC for their time and extensive feedback. We have updated our manuscript with the following main changes:
- Extension of the Control Suite experiments to 9 seeds for all agents (Table 1, Figure 9)
- Evaluation of asymptotic behavior with no changes to the trade-off parameter schedule (Figure 14, Appendix L)
- Evaluation of increasing the ensemble size to 10 particles on Cartpole Sparse, Cheetah, Pendulum, and Walker Walk (Figure 13, Appendix J)
- Evaluation of interleaving sampling from the acquisition or evaluation policy on Cheetah and Walker Walk (Figure 15, Appendix M)
- A brief comparison of changing network initialization schemes on Walker Walk (Figure 16, Appendix N)
- A qualitative comparison with the EDL agent on maze exploration (Figure 17, Appendix O)

Thank you very much for your time and valuable suggestions for improving our paper!

---

### Meta-Review · Area_Chair_NWAn · 2021-08-06

**Recommendation:** Accept (Poster)
**Confidence:** 4

**Metareview:**

This paper extends latent-space model-based RL methods by using the uncertainty of the latent model for exploration. The uncertainty estimates are obtained using an ensemble method.

**Pre-rebuttal:** The largest issue with the paper is the low number of seeds. The variance of the results is high and the significance of the results is questionable. Run at least 10 seeds for these and perform a significance test. Only make items bold in the table that are actually significant.

**Post-rebuttal:**
The proposed method is rather incremental but nevertheless interesting. The reviewers pointed out missing comparisons that were made during the rebuttal. Also, the number of repetitions was increased. There are still a few minor things that need to be updated for the camera-ready version: Make sure the numbers in the abstract match the updated results after all 10 seeds are run. Second, perform a significance test and highlight significant entries in the main table.

Other details can be found in the reviews. All reviewers suggest acceptance and since many of the issues were fixed, I recommend acceptance of the paper.

---

> ### Comment · Area_Chair_NWAn · 2021-08-27
> **Waiting for Authors responses**
>
> Dear Authors,
>
> we got 4 reviews here that have raised questions. I also mentioned concerns about the significance. Please take the opportunity and respond to the authors and upload a revised version.
>
> Your AC

---

> > ### Author Response · Authors · 2021-08-28
> > **Initial revision and additional evaluations**
> >
> > We would like to thank the reviewers and the AC for their time and effort extended towards improving the quality of our paper.
> > We apologize for the delay and have now uploaded an initial revised version of the main manuscript and the appendix.
> > Additional evaluations include the following, and corresponding changes are provided in blue:
> > 1. Extension of the Control Suite experiments to 9 seeds for all agents (Table 1, Figure 9)
> > 2. Evaluation of asymptotic behavior with no changes to the trade-off parameter schedule (Figure 14)
> > 3. Evaluation of increasing the ensemble size to 10 particles on 2 environments (Figure 13)
> > 4. Evaluation of interleaving sampling from the acquisition or evaluation policy on one task (Figure 15)
> >
> > Due to the computational requirements of running many model-based RL experiments, some evaluations are still ongoing.
> > We hope that the remaining evaluations will conclude soon and we will upload individual responses shortly, as well as another improved version.
> > Thank you very much for your time and valuable comments!

---

### Decision · Program_Chairs · 2021-09-13

**Decision:**

Accept (Poster)

**Comment:**

This paper extends latent-space model-based RL methods by using the uncertainty of the latent model for exploration. The uncertainty estimates are obtained using an ensemble method.

**Pre-rebuttal:** The largest issue with the paper is the low number of seeds. The variance of the results is high and the significance of the results is questionable. Run at least 10 seeds for these and perform a significance test. Only make items bold in the table that are actually significant.

**Post-rebuttal:**
The proposed method is rather incremental but nevertheless interesting. The reviewers pointed out missing comparisons that were made during the rebuttal. Also, the number of repetitions was increased. There are still a few minor things that need to be updated for the camera-ready version: Make sure the numbers in the abstract match the updated results after all 10 seeds are run. Second, perform a significance test and highlight significant entries in the main table.

Other details can be found in the reviews. All reviewers suggest acceptance and since many of the issues were fixed, I recommend acceptance of the paper.